# Epidemiological Analysis of the Emergency Vascular Access in Pediatric Trauma Patients: Single-Center Experience of Intravenous, Intraosseous, Central Venous, and Arterial Line Placements

**DOI:** 10.3390/children10030515

**Published:** 2023-03-05

**Authors:** Manuel Florian Struck, Franziska Rost, Thomas Schwarz, Peter Zimmermann, Manuela Siekmeyer, Daniel Gräfe, Sebastian Ebel, Holger Kirsten, Christian Kleber, Martin Lacher, Bernd Donaubauer

**Affiliations:** 1Department of Anesthesiology and Intensive Care Medicine, University Hospital Leipzig, 04103 Leipzig, Germany; 2Department of Anesthesiology, University Medical Center Göttingen, 37075 Göttingen, Germany; 3Department of Pediatric Surgery, University Hospital Leipzig, 04103 Leipzig, Germany; 4Pediatric Intensive Care Unit, Department of Pediatrics, University Hospital Leipzig, 04103 Leipzig, Germany; 5Institute of Pediatric Radiology, University Hospital Leipzig, 04103 Leipzig, Germany; 6Department of Diagnostic and Interventional Radiology, University Hospital Leipzig, 04103 Leipzig, Germany; 7Institute for Medical Statistics, Informatics, and Epidemiology, Medical Faculty, University of Leipzig, 04107 Leipzig, Germany; 8Department of Orthopedics, Traumatology, and Plastic Surgery, University Hospital Leipzig, 04103 Leipzig, Germany

**Keywords:** pediatric major trauma, vascular access, intravenous line, intraosseous access, central venous catheter, arterial line, prehospital, emergency department

## Abstract

Vascular access in severely injured pediatric trauma patients is associated with time-critical circumstances and low incidences, whereas only scarce literature on procedure performance is available. The purpose of this study was to analyze the performance of different vascular access procedures from the first contact at the scene until three hours after admission. Intubated pediatric trauma patients admitted from the scene to a single Level I trauma center between 2008 and 2019 were analyzed regarding intravenous (IV) and intraosseous (IO) accesses, central venous catheterization (CVC) and arterial line placement. Sixty-five children with a median age of 14 years and median injury severity score of 29 points were included, of which 62 (96.6%) underwent successful prehospital IV or IO access by emergency medical service (EMS) physicians, while it failed in two children (3.1%). On emergency department (ED) admission, IV cannulas of prehospital EMS had malfunctions or were dislodged in seven of 55 children (12.7%). IO access was performed in 17 children without complications, and was associated with younger age, higher injury severity and higher mortality. Fifty-two CVC placements (58 attempts) and 55 arterial line placements (59 attempts) were performed in 45 and 52 children, respectively. All CVC and arterial line placements were performed in the ED, operating room (OR) and intensive care unit (ICU). Ten mechanical complications related to CVC placement (17.8%) and seven related to arterial line placement (10.2%) were observed, none of which had outcome-relevant consequences. This case series suggests that mechanical issues of vascular access may frequently occur, underlining the need for special preparedness in prehospital, ED, ICU and OR environments.

## 1. Introduction

Obtaining vascular access in severely injured pediatric trauma patients is a key skill of acute emergency care providers to initiate fluid resuscitation and administer analgesics, anesthetics and vasopressors [1,2,3,4,5,6]. Even in dedicated trauma centers, time-critical circumstances and low incidences of pediatric trauma cases may lead to vascular access-related mechanical issues that may delay treatment and ultimately cause iatrogenic injuries [5,6,7,8,9,10,11]. This is particularly relevant in critically injured children requiring tracheal intubation due to acute cardiopulmonary decompensation, hemorrhagic shock, impaired consciousness and exposure to extreme pain conditions. There is scarce literature on emergency vascular access in pediatric trauma patients [12].

The purpose of this study was to analyze the performance of different vascular access procedures from the first contact at the scene until three hours after admission to a Level I trauma center. The primary endpoints were to identify frequencies and performances of intravenous access (IV), intraosseous access (IO), central venous catheterization (CVC) and arterial line placement related to access sites in prehospital emergency medical service (EMS), emergency department resuscitation room (ED), operating room (OR) and intensive care unit (ICU) locations.

## 2. Materials and Methods

### 2.1. Study Design

A dataset of consecutive acute pediatric major trauma patients requiring emergency tracheal intubation was reanalyzed regarding vascular access [13]. Ethical approval and retrospective registration were obtained by the Ethics Committee of the Medical Faculty of Leipzig, Germany (No. 441/15-ek) and the German Clinical Trials Register (DRKS00028045).

The study sample included major trauma pediatric patients in the local trauma registry of the University Hospital Leipzig admitted between 2008 and 2019 (the raw data can be found in Appendix A). The inclusion criteria were <18 years of age, with admission directly from the scene and emergency tracheal intubation at the scene or in the emergency department. Data were obtained from paper-based and computerized charts (PDMS COPRA 5, COPRA System GmbH, Berlin, Germany), the radiological information system and the picture archiving and communication system.

### 2.2. Vascular Access Complications

The vascular access-related mechanical complications were classified according to the Common Terminology Criteria for Adverse Events (version 5.0). Bleeding Grade I and II, arrhythmia Grades I and II, inadvertent arterial puncture, nonpersistent nerve injury, failed catheterization, secondary dislodgements, functionality issues and CVC tip malposition were classified as minor mechanical complications. Bleeding Grades III and IV, arrhythmia Grades III and IV, inadvertent arterial catheterization, limb ischemia, persistent nerve injury and pneumothorax were classified as major mechanical complications.

### 2.3. General Management in the Study Center

The prehospital treatment of pediatric major trauma patients is performed by emergency response physicians until hospital admission. Peripheral IV and IO access devices are available on all physician-staffed rapid response units and two EMS helicopters. Resuscitation room activation and management are organized according to the recommendations of the German Society of Trauma Surgery (DGU) and the German Society of Pediatric Surgery (DGKCH). The interdisciplinary trauma team performs a standardized assessment, whereas intensivists and anesthetists are responsible for IV, IO, CVC and arterial line placement.

### 2.4. Statistical Analysis

The data are presented as numbers (percentages), whereas the normally distributed data are presented as the mean ± standard difference (SD) and the nonnormally distributed data are presented as the median (interquartile range (IQR)). Normal distributions were tested via the Kolmogorov–Smirnov test. Patient characteristics were compared by applying univariable logistic regression analysis or Firth’s bias-reduced logistic regression according to the number of observations. The investigated risk factors for vascular access-related complications were age, sex, weight, Glasgow coma scale (GCS) score, injury severity score (ISS) and performance of cardiopulmonary resuscitation (CPR). The investigated outcome factors were 24-h mortality and 30-day mortality. The alpha level of significance was set at 0.05. To control for the alpha error, we used the Holm’s method. All tests were two-tailed. Multivariable analysis was not performed due to the difficult comparability of different procedures and locations. All analyses were performed in the framework of R 4.2.2.

## 3. Results

### 3.1. Baseline Characteristics

Sixty-five children were analyzed (40 (61.5%) male, median age 14.0 (8.6) years) (Table 1). The age distribution was as follows: age <1 year, no patient; 1–5 years, 12 patients (18.5%); 6–11 years, 14 patients (21.5%); and 12–17 years, 39 patients (60%). The median ISS was 29 (21) points. The 24-h mortality was 12.3% (n = 8) and the 30-day mortality was 27.7% (n = 18). Nineteen children (29%) arrived under CPR or had a return of spontaneous circulation at ED admission after prehospital cardiac arrest, and 45 children (69.2%) underwent emergency surgery before ICU admission.

### 3.2. Intravenous Line Placement and Intraosseous Access

Overall, all 65 children received a total of 153 IV line placements. In the prehospital setting, successful IV access was performed in 55 children (84.6%), among which 26 received one IV cannula, 27 received two cannulas and two received three cannulas (Table 2 and Table 3). EMS intraosseous access was performed in 11 children at the scene, of which three received additional prehospital IV access after IO access (Table 2 and Table 4). All but two children (96.6%) had functional IV or IO access at ED admission.

After admission to the resuscitation room, placement or replacement of IV lines was performed in 25 cases (38.5%) without complications. IO access was established in six children, including four with dislodged/malfunctioning IV cannulas placed by prehospital EMS and two children who arrived without IV/IO access. All children in the study cohort had functional vascular access during resuscitation room treatment. Overall, 17 children (26.1%) received IO access, which was associated with younger age, higher injury severity and worse outcome compared with children without IO access (Table 5).

In the OR, 13 children received additional peripheral IV access in the OR (nine with one new IV line and four with two new IV lines), whereas no IO access was performed.

Of the 17 children who were admitted to the ICU without undergoing emergency surgery, seven received one new additional IV line in the ICU, whereas no IO access was performed in the ICU. Forty-four children were admitted postoperatively to the ICU (one child died in the OR), among which no new peripheral IV line placement or IO access was performed.

#### Mechanical Complications Related to IV and IO Access

Due to incomplete or missing documentation, the number of prehospital puncture attempts could not be analyzed. In two children (2 and 15 years, 3.1%), neither IV nor IO access could be established, and they were rushed to the ED under CPR, where vascular access was obtained in both cases. Upon ED admission, IV cannulas placed by prehospital EMS had malfunctioned or were dislodged in seven children (12.7%), including one with *right external jugular venous* access. All IV-related issues were minor complications. IO access was not associated with functionality issues or other mechanical complications.

### 3.3. Central Venous Catheter Placement

Overall, 45 children (69.2%) received 52 CVC placements after 58 puncture attempts during acute care treatment (Table 2 and Table 6).

In the emergency department, central venous access was attempted in 28 children and performed in 26 children, 13 via the right subclavian vein (50%), five via the left subclavian vein (19.2%), four via the right femoral vein (15.4%) and four via the left femoral vein (15.4%).

In the operating room, 16 CVC placements were performed after 20 puncture attempts in 16 children, three of whom had already undergone CVC placement in the ED (including one with misplacement and two for additional capacity). Four children received CVC via the right internal jugular vein (25.0%), one received CVC via the left internal jugular vein (6.2%), seven received CVC via the right femoral vein (43.8%) and four received CVC via the left femoral vein (25.0%). In summary, 35 children underwent surgery with a CVC either placed in the ED or in the OR, and 10 children underwent surgery without CVC placement.

Children admitted to the ICU without undergoing emergency surgery had already undergone CVC placement in the ED in seven cases, and six children received CVC in the ICU (including two with an ED CVC in place). One child received CVC via the left internal jugular vein, four via the right femoral vein and one via the left femoral vein. In summary, 11 children who were admitted to the ICU without emergency surgery had a CVC either placed in the ED or in the ICU, and six children did not receive a CVC. Of the 45 children who underwent emergency surgery, four received a CVC postoperatively in the ICU (one via the right internal jugular vein, two via the right femoral vein and one via the left femoral vein).

#### Mechanical Complications Related to CVC Placement

Overall, ten mechanical complications were observed in 58 CVC puncture attempts (17.2%) in seven children (including one with two complications and one with three complications), all of which were minor. Iatrogenic pneumothorax or major bleedings were not observed. CVC complications were not associated with demographic variables, injury severity or outcome (Table 7).

### 3.4. Arterial Line Placement

Overall, 52 children (80%) received 55 arterial line placements after 59 puncture attempts during acute care treatment (Table 2 and Table 8).

In the ED, arterial line placement was attempted in 29 children and performed in 28 children, six of via the right radial artery (21.4%), seven via the left radial artery (25%), 10 via the right femoral artery (35.7%) and five via the left femoral artery (17.8%).

In the OR, arterial line placement was performed in 22 children, including two who had undergone ED arterial line placement (one for improved access to the right radial artery instead of the right femoral artery and one who required endovascular bleeding control via a right femoral artery sheath in addition to the existing right radial artery line). Five children (22.7%) received arterial lines via the right radial artery, four (18.2%) via the left radial artery, nine (40.9%) via the right femoral artery and three (13.6%) via the left femoral artery. In summary, 39 children (86.7%) underwent surgery with an arterial line either placed in the ED or in the OR, and six children (13.3%) underwent surgery without an arterial line.

In the ICU, arterial line placement in the 17 children without emergency surgery was performed in four cases in the ICU (23.5%, three via the right radial arteries and one via the left femoral artery), whereas a total of 12 children (70.6%) had an arterial line either placed in the ED or in the ICU, and five children (29.4%) did not receive an arterial line (including one with CVC placement). Among the postoperatively admitted children, one received arterial line replacement.

#### Mechanical Complications Related to Arterial Line Placement

Overall, six mechanical complications in 59 arterial line puncture attempts (10.2%) were observed, which included failed accesses and functionality issues (minor complications) and one case of suspected limb ischemia (major complication). Arterial line placement-related issues were not associated with demographic variables, injury severity or outcomes (Table 9).

## 4. Discussion

In our study cohort, most mechanical complications of vascular access procedures were minor, and were not associated with demographic variables, injury severity or outcomes. In the literature, there are no studies available that analyzed the vascular access performance of the entire acute care phase of pediatric trauma patients. The comparability of existing studies on pediatric vascular access is difficult due to considerably different settings, designs, sample sizes and groups of operators [6,8,9,12,14,15,16,17,18,19,20,21,22,23,24,25].

At the scene, the prehospital EMS team is the first contact of healthcare professionals with the injured child. Rapid IV access for the immediate treatment of severe shock and cardiac arrest or the induction of emergency anesthesia is paramount [1,2]. Due to reduced venous filling and impaired dermal perfusion, multiple puncture attempts are common [2,4,5,6]. To avoid a delay of lifesaving treatment, current guidelines recommend puncture success within two minutes, and if not successful, immediately attempt IO access [1].

In our study, we could not analyze puncture approaches in the prehospital setting in detail due to documentation issues or missing data. However, 84.6% of patients received successful prehospital IV access, whereas 96.9% had prehospital IV or IO access. Other studies found overall success rates of prehospital IV access in 77%, 84% and 88.3% of cases [12,13,14,15,16]. First puncture attempt success rates in the literature range from 53% [17] and 67.8% [18] to 73% and 81% [19]. Even in nonemergency environments of elective OR settings, up to 9% of cases need more than three attempts to obtain successful IV cannulation [20]. Although easily obtainable patient characteristics may not be appropriate to precisely identify the difficulty of IV access [19], most studies report younger age as a risk factor [15,16,18,19,20,21,22,23]. The difficult intravenous access score (DIVA) includes age and vein appearance as predictive variables, and has further been modified since its first description (3 points for prematurity and for age under 1 year, 2 points for vein not palpable and for vein not visible, 1 point for 1–2 years of age) [18,23,24]. Robust factors that may improve IV success rates are experienced operators [25] and the use of ultrasound [26], whereas the use of infrared light devices depends on patient selection and operator training [27,28]. In our study, a considerable number of new IV accesses or replacements were performed in the ED, which may reflect the improved infrastructure of a standardized resuscitation room environment and underlines the importance of a structured first survey under clinical conditions.

Our results are in line with other observations that IO access is a surrogate of the critical condition and the need for CPR [29,30]. An analysis of a French database, however, found that IO access or peripheral IV access did not affect survival after pediatric out-of-hospital cardiac arrest [31]. The availability of provider-friendly drilling devices, and thus the opportunity for immediate IO access, may contribute to the increasing confidence of acute care providers, lower threshold to use and faster administration of drugs, particularly in patients under cardiac arrest or acute life-threatening conditions [11,32]. However, two recent studies reported considerably different rates of mechanical complications and overall successes, probably because of different operator training and experience [33,34]. The most common anatomical IO access site in our study was the proximal tibia (anteromedial aspect, 2–3 cm below the tibia tuberosity), which was used in all but one case (which used the proximal humerus). Although we observed no mechanical complications, the choice of the IO needle size and the puncture level may affect the needle tip position, which is particularly relevant considering the anatomy of small children under 2 years [35,36].

For fluid resuscitation, including transfusion of blood products, the flow rates of IO accesses are limited, and at least two peripheral IV lines are recommended, while the size of the cannulas should be large enough to deliver appropriate volumes rapidly [1,2,5,6,37]. Recent studies revealed that continuous administration of vasopressor agents through peripheral IV lines is feasible and safe [38,39,40]. However, special attention should be paid to the patency, functionality and securement of peripheral IV lines to avoid extravasation injuries.

When peripheral IV and IO access is not sufficient, CVC placement is indicated. In our study cohort, over two-thirds of patients received a CVC, half of whom were in the ED, and the other half having one placed later in the OR or ICU. CVC-related mechanical complications occurred in 17.2% of puncture attempts (15.5% of patients), which is in line with the results of previous studies ranging from 1.3% [41], 3.3% [42], 12.8% [43], 14.1% [44], 20.2% [45] 25% and 31% [46] up to 38.4% [47]. Mechanical complications and malposition had no relevant impact on the treatment course, supporting other studies in the pediatric population and adult trauma population of our study center, in which we found considerable malposition rates of almost 30% [47,48]. In the literature, common factors associated with mechanical complications in pediatric CVC placement are left-sided catheters, patient weight, sex, the team´s learning curve and the use of ultrasound [9,41,42,43,44,47]. The emergency placement of a CVC in pediatric patients requires considerable experience and training under nonemergency circumstances, and should not be performed by novices [11,49]. In small children, the femoral access route is preferable due to less interaction with other procedures performed on the upper body part (e.g., CPR) and low complication potential compared to subclavian and internal jugular approaches [2,42]. Furthermore, cervical spine immobilization usually does not allow unrestricted jugular access in trauma patients.

In our study cohort, two-thirds of ED CVC placements were performed in the subclavian veins, whereas one-third were performed in the femoral veins. This probably reflects the age categories of our patients, in which the majority were adolescents. Most CVC placements after the acute resuscitation phase in the OR and the ICU were performed in the femoral veins, while jugular approaches were performed after radiological confirmation of the absence of cervical spine injuries. Although subclavian CVC has been proposed as a safe procedure, it carries the risk of iatrogenic arterial injuries and pneumothorax, particularly in patients on mechanical ventilation [44]. In this context, it should be acknowledged that CVC placement via the subclavian vein should be performed on the ipsilateral side of the chest tube insertion in case thoracic injuries are present. Although we only performed ultrasound-guided CVC placement in small children, compared with landmark-guided approaches in adolescents, there is increasing evidence that the general use of ultrasound improves performance success and reduces complication rates [50,51,52,53].

In severely injured patients on mechanical ventilation, the placement of the arterial line allows for blood gas analysis and invasive blood pressure monitoring. Eighty percent of the children in our study cohort received arterial line placement, whereas the remaining patients presented either less severe injuries within a very short mechanical ventilation time or were under CPR at admission and died in the ED or the OR. The most common access sites were the radial and femoral arteries, with a mechanical complication rate of 10.2%. One case of suspected limb ischemia after femoral artery catheterization was the only major complication of the study, which highlights the need of special attention. Previous studies on pediatric arterial line placement revealed complication rates of 0.2% [54] and 33% [55]. Similar to CVC placement, ultrasound use is increasingly recommended to improve first pass success and reduce complications [55].

### Limitations

The general limitations of retrospective studies, small sample sizes and a single center design are acknowledged. We present a special cohort of critically injured patients requiring tracheal intubation and other patients receiving no advanced airway management, which may present different vascular access characteristics, complication rates and outcomes. Age classes were not equally distributed in the study sample, and adolescents were overrepresented compared to small children. Detailed analysis of the IV access performance (i.e., multiple puncture approaches) was not possible due to inconsistent documentation and/or missing data. We also cannot exclude underreporting of complications of IO access, CVC and arterial line placements due to the emergency situations in some cases. The exploratory character of the analysis implicates the necessity for replication, as correcting for multiple testing was done per outcome, and the size of the dataset was moderate. Nevertheless, we present comprehensive data covering the different vascular access performances from the prehospital setting until the first three hours of in-hospital acute care treatment.

## 5. Conclusions

In this case series, emergency vascular access procedures in severely injured children were performed in different locations, and mechanical complications had no impact on outcomes. Intraosseous access is associated with a critical condition and young age, and is exclusively used in the acute resuscitation phase. Common central venous catheter insertion sites are femoral veins in younger children and subclavian veins in adolescents. Prospective studies with appropriate sample sizes over all pediatric age classes are required to evaluate the performance quality of emergency vascular access in pediatric trauma care.

## Figures and Tables

**Table 1 children-10-00515-t001:** Baseline characteristics of the study cohort.

Variable	
Age (years)	14.0 (8.5)
Male, *n* (%)	40 (62)
Weight (kg)	51.2 ± 25.1
GCS	8 (8)
ISS	29 (21)
CPR before ED, *n* (%)	19 (29.2)
24-h mortality, *n* (%)	8 (12.3)
30-d mortality, *n* (%)	18 (27.7)

Median (interquartile range); mean ± standard deviation; ISS, injury severity score; CPR, cardiopulmonary resuscitation; ED, emergency department.

**Table 2 children-10-00515-t002:** Overview on vascular accesses related to performance locations.

Procedure	EMS	ED	OR	ICU
Intravenous line placement	55	26	13	7
Intraosseous access	11	6	0	0
Central venous catheter placement	0	26	16	10
Arterial line placement	0	28	22	5

EMS, emergency medical service; ED, emergency department; OR, operating room; ICU, intensive care unit.

**Table 3 children-10-00515-t003:** Intravenous line placement.

Number of Intravenous Lines	EMS	ED	OR	ICU
1	26	23	9	7
2	27	1	4	0
3	2	2	0	0

EMS, emergency medical service; ED, emergency department; OR, operating room; ICU, intensive care unit. Numbers are new placements in each location.

**Table 4 children-10-00515-t004:** Intraosseous access sites.

Access Site	EMS	ED	OR	ICU
Right proximal tibia	7	4	0	0
Left proximal tibia	3	2	0	0
Right proximal humerus	0	0	0	0
Left proximal humerus	1	0	0	0

EMS, emergency medical service; ED, emergency department; OR, operating room; ICU, intensive care unit.

**Table 5 children-10-00515-t005:** Comparison of patients with and without intraosseous access.

Variable	No IO Access	IO Access	*p*	Adjusted *p*
Age (years)	15 (6.75)	8 (14)	0.006	0.018
Male, *n* (%)	32 (66.6)	8 (47)	0.157	0.157
Weight (kg)	60 (30)	25 (48.5)	0.013	0.027
GCS	10 (9)	3 (1.5)	0.001	0.004
ISS	25 (17)	45 (37.5)	<0.001	0.003
CPR before ED, *n* (%)	6 (12.5)	13 (76.5)	<0.001	<0.001
24-h mortality, *n* (%)	1 (2.1)	7 (41.2)	<0.001	0.004
30-d mortality, *n* (%)	5 (10.4)	13 (76.5)	<0.001	<0.001

Median (interquartile range); mean ± standard deviation; IO, intraosseous; GCS, Glasgow Coma Scale; ISS, injury severity score; CPR, cardiopulmonary resuscitation; ED, emergency department.

**Table 6 children-10-00515-t006:** Central venous catheter placement sites.

Vein	EMS	ED	OR	ICU
Right internal jugular	0	0	4	1
Left internal jugular	0	0	1	1
Right subclavian	0	13	0	0
Left subclavian	0	5	0	0
Right femoral	0	4	7	6
Left femoral	0	4	4	2

EMS, emergency medical service; ED, emergency department; OR, operating room; ICU, intensive care unit.

**Table 7 children-10-00515-t007:** Comparison of patients with and without CVC-related mechanical complications.

Variable	No Complications (*n* = 38)	Complications (*n* = 7)	*p*
Age (years)	15 (7)	15 (4)	0.545
Male, *n* (%)	25 (65.8)	4 (57.1)	0.352
Weight (kg)	52.4 ± 30.8	59.9 ± 32.8	0.457
GCS	5 (7)	3 (4)	0.337
ISS	32 (18)	44 (38.5)	0.238
CPR before ED, *n* (%)	14 (36.8)	5 (71.4)	0.210
24-h mortality, *n* (%)	5 (13.1)	3 (42.8)	0.123
30-d mortality, *n* (%)	13 (34.2)	5 (71.4)	0.163

Median (interquartile range); mean ± standard deviation; IO, intraosseous; GCS, Glasgow Coma Scale; ISS, injury severity score; CPR, cardiopulmonary resuscitation; ED, emergency department.

**Table 8 children-10-00515-t008:** Arterial line placement sites.

Artery	EMS	ED	OR	ICU
Right radial	0	6	5	3
Left radial	0	7	4	1
Right femoral	0	10	9	0
Left femoral	0	5	3	1

EMS, emergency medical service; ED, emergency department; OR, operating room; ICU, intensive care unit.

**Table 9 children-10-00515-t009:** Comparison of patients with and without arterial line-related mechanical complications.

Variable	No Complications (*n* = 46)	Complications (*n* = 6)	*p*
Age (years)	15 (9)	13.5 (3)	0.867
Male, *n* (%)	29 (63.0)	3 (50.0)	0.540
Weight (kg)	60.0 ± 25.8	51.2 ± 25.1	0.820
GCS	6 (7)	12 (8)	0.093
ISS	29 (20)	35 (29)	0.545
CPR before ED, *n* (%)	15 (32.6)	1 (16.7)	0.438
24-h mortality, *n* (%)	6 (13.0)	0 (0)	0.599
30-d mortality, *n* (%)	13 (28.2)	2 (33.3)	0.796

Median (interquartile range); mean ± standard deviation; IO, intraosseous; GCS, Glasgow Coma Scale; ISS, injury severity score; CPR, cardiopulmonary resuscitation; ED, emergency department.

## Data Availability

Data supporting the reported results can be found in Appendix A: RAW_DATA_FILE.

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
