# Peer review of "Epidemiological Analysis of the Emergency Vascular Access in Pediatric Trauma Patients: Single-Center Experience of Intravenous, Intraosseous, Central Venous, and Arterial Line Placements"

_children, 2023, doi:10.3390/children10030515_

Round 1

Reviewer 1 Report

Congratulations to the paper, interesting and important issue, in general well written.

The statistical tests need correction of the alpha error accumulation!!! (24 tests/p-values without any correction is not acceptable) Therefore p-value 1 and 3 in table 5, which do not withstand correction are no longer significant. (I recommend to minimize the applied test to a minimum - since p-values  in table 7 and 9 are not significant anyway). This section/issue needs to be redone in accordance to a statistician.

Author Response

Reviewer 1

Congratulations to the paper, interesting and important issue, in general well written.

The statistical tests need correction of the alpha error accumulation!!! (24 tests/p-values without any correction is not acceptable) Therefore p-value 1 and 3 in table 5, which do not withstand correction are no longer significant. (I recommend to minimize the applied test to a minimum - since p-values  in table 7 and 9 are not significant anyway). This section/issue needs to be redone in accordance to a statistician.

RESPONSE: We would like to thank the reviewer for this important point and agree that a correction of the alpha error is required in multiple testing.

We re-calculated the dataset using univariable logistic regression analysis and Firth's bias-reduced logistic regression according to the number of observations instead of the simple Mann-Whitney-U tests and Fisher-Exact tests. P-values in the tables 5, 7 and 9 showed similar significance levels as in the previous calculations.

To control for the alpha error, we used the Holm method, which may be superior compared with the Bonferroni method.

  1. Aickin M, Gensler H. Adjusting for multiple testing when reporting research results: the Bonferroni vs Holm methods. Am J Public Health. 1996 May;86(5):726-8. doi: 10.2105/ajph.86.5.726.
  2. Levin B. On the Holm, Simes, Hochberg multiple test procedures. Am J Public Health. 1996 May;86(5):628-9. doi: 10.2105/ajph.86.5.628.

In table 5, p-value 1 (age) remained significant when using the Holm method (p=0.0186) and the Bonferroni method (p=0.049, whereas p-value 3 (body weight) remained significant when using the Holm method (p=0.027) but not using the Bonferroni method (p=0.108).

If the reviewer prefers the Bonferroni method for alpha error correction, we surely can present these values instead. 

We have added to the methods section: “Patient characteristics were compared by applying univariable logistic regression analysis or Firth's bias-reduced logistic regression according to the number of observations”. And  “To control for the alpha error, we used the Holm´s method.”

Furthermore, we have added in the limitations section: “The exploratory character of the analysis implicates the necessity for replication as correcting for multiple testing was done per-outcome and the size of the dataset was moderate.”

We would like to thank the reviewer for his valuable comments. We completely reworked the calculations including the alpha error correction, thanks to our co-author and statistician Dr. Holger Kirsten. The manuscript underwent professional language editing (see AJE certificate) and another processing by a native speaking colleague.

We hope that we have addressed all items appropriately and thank the reviewer again for his time.

Reviewer 2 Report

Dear authors,

Thank you very much for your effort in providing this article.

This article gives an epidemiological insight on the emergency vascular access in pediatric patients.

Here below I’m providing you with some suggestions and remarks to improve the quality of your work.

Wish you luck in revising the current version

Best regards

Reviewer

 Please modify the title

The wording “intubated” does not add extra information about this article.

A more appropriate title could be “epidemiological analysis of the emergency vascular access in pediatric trauma patients: …. “

I’m not sure about the reliability of the data as they go back to 2008 when computerized charts were only limited available globally and surely not at the out of hospital setting. Also the existence of an accurate registration of data about the numbers of attempts in medical file is questionable surely on the time of CPR, out of hospital and in stressful situations when dealing with an acute and seriously injured child. How can the authors guarantee the accuracy of their data?

Is it really interesting to mention after how many attempts was someone successful to place an iv line in a child in resuscitation setting, definitely if this did not affect the outcome?  

I cannot understand the purpose of this manuscript that provides a nice epidemiological description on the emergency vascular access in children

Line 31.

I cannot fully understand this sentence “Among the 65 children included (median age 14.0 years, and median ISS 29 points), 62 (96.9%)” please modify the wording.

Please fully write ISS in the abstract to understand the text better

Line 40.

There are no results provided in this article with respect to the ICU or OR environment and conditions. However, in the conclusion it is stated that “This case series suggests that mechanical issues of vascular access may frequently occur, underlining the need for special preparedness in prehospital, ED, ICU, and OR environments”. This should also mirror the results discussed in the abstract. It would be sensible that the authors add this information to the results or delete them from the conclusion.

Author Response

Reviewer 2

Dear authors,

Thank you very much for your effort in providing this article.

This article gives an epidemiological insight on the emergency vascular access in pediatric patients.

Here below I’m providing you with some suggestions and remarks to improve the quality of your work.

Wish you luck in revising the current version

Best regards

Reviewer

Please modify the title

The wording “intubated” does not add extra information about this article.

A more appropriate title could be “epidemiological analysis of the emergency vascular access in pediatric trauma patients: …. “

RESPONSE: We agree with the reviewer and changed the title accordingly.

I’m not sure about the reliability of the data as they go back to 2008 when computerized charts were only limited available globally and surely not at the out of hospital setting. Also the existence of an accurate registration of data about the numbers of attempts in medical file is questionable surely on the time of CPR, out of hospital and in stressful situations when dealing with an acute and seriously injured child. How can the authors guarantee the accuracy of their data?

RESPONSE: We would like to thank the reviewer for this comment. The University Hospital Leipzig has introduced the digital health record COPRA electronic documentation system for all ED, OR, and ICU patients charts in 2001. Thus, we can confirm that all information given in these digital charts were used in this analysis. However, we cannot exclude underreporting of complications due to the emergency situation in some cases. For clarity, we have added two more sentences in the methods and limitations section: “Data were obtained from paper-based and computerized charts (PDMS COPRA 5, COPRA System GmbH, Berlin, Germany), the radiological information system, and the picture archiving and communication system.” And “We also cannot exclude underreporting of complications of IO access, CVC and arterial line placements due to the emergency situations in some cases.”

Is it really interesting to mention after how many attempts was someone successful to place an iv line in a child in resuscitation setting, definitely if this did not affect the outcome? 

I cannot understand the purpose of this manuscript that provides a nice epidemiological description on the emergency vascular access in children

Line 31.

RESPONSE: Our purpose to conduct this analysis was to reflect all vascular accesses in pediatric major trauma patients from the first approach at the scene until three hours of admission to the trauma center. To our knowledge, comprehensive data that evaluates IV, IO, CVC, and arterial lines and their mechanical complications in the acute resuscitation phase are not available in the literature. Thus, we believe that our manuscript may serve as an idea for further prospective studies with more appropriate sample sizes across all pediatric age classes. 

I cannot fully understand this sentence “Among the 65 children included (median age 14.0 years, and median ISS 29 points), 62 (96.9%)” please modify the wording.

RESPONSE: We have modified this sentence to: “Sixty-five children with a median age of 14 years and median injury severity score of 29 points were included, of which 62 (96.6%) underwent successful prehospital IV or IO access by...”

Please fully write ISS in the abstract to understand the text better

Line 40.

RESPONSE: We have written “injury severity score” instead of the abbreviation, accordingly.

There are no results provided in this article with respect to the ICU or OR environment and conditions. However, in the conclusion it is stated that “This case series suggests that mechanical issues of vascular access may frequently occur, underlining the need for special preparedness in prehospital, ED, ICU, and OR environments”. This should also mirror the results discussed in the abstract. It would be sensible that the authors add this information to the results or delete them from the conclusion.

RESPONSE: Thank you for this point. We analyzed all vascular accesses in pediatric major trauma patients until three hours after admission to our trauma center. This included EMS, ED, OR, and ICU environments. In tables 2, 3, 4, 6, and 8, we have listed the OR and ICU characteristics of the patients.

For better understanding, we have added to the abstract: “All CVC and arterial line placements were performed in the ED, OR and ICU.”

We would like to thank the reviewer for his valuable comments. We hope that we have addressed all items appropriately. Furthermore, we have re-calculated Tables 5, 7, and 9 according to reviewer1 sugestions, which resulted in similar findings and did not change the main conclusions.

We feel that the manuscript has improved considerably and thank the reviewer again for his time.

Round 2

Reviewer 1 Report

if statistics were redone with a statistician changes are fine for me

Reviewer 2 Report

Dear authors, 

Thank you very much for providing this revised version that is much more readable now

My questions are very well answered and I don't have further suggestions. I can approve this version for publication; Congratulations

Best regards

Reviewer